# Evaluation of Structure Prediction and Molecular Docking Tools for Therapeutic Peptides in Clinical Use and Trials Targeting Coronary Artery Disease

**DOI:** 10.3390/ijms26020462

**Published:** 2025-01-08

**Authors:** Nasser Alotaiq, Doni Dermawan

**Affiliations:** 1Health Sciences Research Center (HSRC), Imam Mohammad Ibn Saud Islamic University (IMSIU), Riyadh 13317, Saudi Arabia; 2Department of Applied Biotechnology, Faculty of Chemistry, Warsaw University of Technology, 00-661 Warsaw, Poland; doni.dermawan.stud@pw.edu.pl

**Keywords:** apelin, coronary artery disease, molecular docking, molecular dynamics, structure prediction, therapeutic peptides

## Abstract

This study evaluates the performance of various structure prediction tools and molecular docking platforms for therapeutic peptides targeting coronary artery disease (CAD). Structure prediction tools, including AlphaFold 3, I-TASSER 5.1, and PEP-FOLD 4, were employed to generate accurate peptide conformations. These methods, ranging from deep-learning-based (AlphaFold) to template-based (I-TASSER 5.1) and fragment-based (PEP-FOLD), were selected for their proven capabilities in predicting reliable structures. Molecular docking was conducted using four platforms (HADDOCK 2.4, HPEPDOCK 2.0, ClusPro 2.0, and HawDock 2.0) to assess binding affinities and interactions. A 100 ns molecular dynamics (MD) simulation was performed to evaluate the stability of the peptide–receptor complexes, along with Molecular Mechanics/Poisson–Boltzmann Surface Area (MM/PBSA) calculations to determine binding free energies. The results demonstrated that Apelin, a therapeutic peptide, exhibited superior binding affinities and stability across all platforms, making it a promising candidate for CAD therapy. Apelin’s interactions with key receptors involved in cardiovascular health were notably stronger and more stable compared to the other peptides tested. These findings underscore the importance of integrating advanced computational tools for peptide design and evaluation, offering valuable insights for future therapeutic applications in CAD. Future work should focus on in vivo validation and combination therapies to fully explore the clinical potential of these therapeutic peptides.

## 1. Introduction

Coronary artery disease (CAD) remains one of the leading causes of mortality worldwide, characterized by the narrowing or blockage of coronary arteries due to atherosclerosis [1,2]. This condition results in reduced blood flow to the heart, leading to angina, myocardial infarction, and heart failure [3]. Current treatment options for CAD include lifestyle modifications, pharmacological interventions, and invasive procedures such as angioplasty and coronary artery bypass grafting. However, these treatments often address symptoms rather than the underlying molecular mechanisms driving the disease [4,5]. Consequently, there is an urgent need to develop novel therapeutics that can effectively target the key pathways involved in CAD progression. Peptide-based therapeutics have emerged as a promising class of drugs due to their high specificity, low immunogenicity, and ability to modulate protein–protein interactions [6,7]. Therapeutic peptides can potentially target the critical pathways associated with CAD, including inflammation, endothelial dysfunction, oxidative stress, and lipid metabolism [8,9]. Peptides such as glucagon-like peptide-1 (GLP-1) analogs, Nesiritide, and Apelin are either approved for clinical use or undergoing trials for CAD treatment, demonstrating their relevance as effective therapeutic agents [10,11]. Despite this promise, the design and optimization of peptide therapeutics remain challenging due to the need for accurate structural modeling and interaction prediction with specific target receptors [12,13].

Computational tools have revolutionized drug discovery by enabling in silico design and evaluation of therapeutic candidates [14,15]. For peptides, two critical steps are structure prediction and molecular docking. Structure prediction determines peptides’ three-dimensional (3D) conformation, essential for understanding their functional properties and interaction mechanisms. Molecular docking simulates the binding of peptides to target receptors, providing insights into binding affinity, specificity, and potential interaction sites [16,17,18]. The accuracy and reliability of these tools directly impact the efficiency of peptide drug development. To address these challenges, this study aimed to systematically evaluate web-based computational tools for peptide structure prediction and molecular docking, focusing on their application to therapeutic peptides targeting CAD. The structure prediction tools included AlphaFold 3, I-TASSER 5.1, and PEP-FOLD 4, representing a spectrum of methodologies from deep-learning-based predictions (AlphaFold) to template-based modeling (I-TASSER) and fragment-based de novo predictions (PEP-FOLD). These tools were selected for their availability, accuracy, and proven capabilities in predicting reliable peptide conformations. For molecular docking, four web-based platforms were assessed: HADDOCK 2.4, HPEPDOCK 2.0, ClusPro 2.0, and HawDock 2.0. These tools encompassed a range of docking methodologies, including rigid-body docking, flexible docking, and physics-based scoring functions. Each tool was evaluated for its usability, computational efficiency, and ability to predict accurate peptide–receptor binding modes. By focusing on tools available in web-based formats, the study ensured accessibility and relevance for researchers without requiring extensive computational resources.

The study applied these tools to a set of clinically relevant therapeutic peptides, including FX06, Liraglutide, Exenatide, Nesiritide, Apelin, GLP-1, and Atrial Natriuretic Peptide (ANP). These peptides target receptors implicated in CAD pathophysiology, such as angiotensin II type 1 receptor (AT1R), glucagon-like peptide-1 receptor (GLP-1R), and interleukin-6 receptor (IL-6R). Target receptors were selected based on their direct involvement in vascular remodeling, inflammatory response, and cardiac function pathways. Molecular dynamics (MD) simulations were performed using GROMACS to validate docking results and ensure realistic interaction modeling. MD simulations accounted for the dynamic nature of peptide–receptor complexes under physiological conditions, providing a robust framework for evaluating therapeutic peptides’ stability and binding interactions. This study aimed to provide a comprehensive framework for peptide-based drug design targeting CAD by systematically comparing these tools and integrating MD validation. The findings offered valuable insights into the strengths and limitations of available computational platforms, contributing to optimizing peptide therapeutics for cardiovascular diseases and advancing their translation from bench to bedside.

## 2. Results

### 2.1. Evaluation of Structure Prediction Tools

The performance of three widely used structure prediction tools (AlphaFold 3, I-TASSER 5.1, and PEP-FOLD 4) was systematically evaluated for their ability to generate accurate 3D models of therapeutic peptides. The assessment focused on Z-scores, Ramachandran plot analyses, overall model quality, and local model quality to determine the most reliable tool for peptide structure prediction (Table 1). Among the tools, AlphaFold 3 consistently demonstrated superior performance, producing models with the most favorable Z-scores and minimal outliers in the Ramachandran plots, making it the optimal choice for predicting peptide structures. The complete results can be seen in Appendix A.

The Z-score, a critical metric for assessing predicted models’ structural reliability and statistical quality, highlighted clear distinctions between the tools [19,20]. As the first test case, Apelin exhibited a Z-score of −4.21 when predicted by AlphaFold 3, indicating a high-quality model. In comparison, the Z-score for I-TASSER 5.1 was less favorable at −2.06, while PEP-FOLD 4 produced a significantly poorer Z-score of −1.15. This marked difference underscores AlphaFold 3’s capability to generate statistically robust and structurally accurate models. This trend was observed across all tested peptides. For example, FX06, another therapeutic peptide, had a Z-score of −4.72 with AlphaFold 3, outperforming I-TASSER 5.1 (−4.46) and PEP-FOLD 4, which generated a much less favorable Z-score of 0.11. Similarly, for GLP-1, AlphaFold 3 yielded a Z-score of −0.95, slightly better than I-TASSER 5.1 (−0.91) and considerably better than PEP-FOLD 4 (−0.72). These results establish AlphaFold 3 as the most reliable tool in terms of Z-score evaluation, especially for peptides with complex secondary and tertiary structures. Ramachandran plot validation further demonstrated AlphaFold 3’s superior structural accuracy. For all peptides, the models generated by AlphaFold 3 exhibited the fewest outliers in the disallowed regions, indicating proper backbone dihedral angles and minimal steric clashes. In the case of Apelin, AlphaFold 3 achieved an optimal distribution of residues in the favored and allowed regions, with only a negligible number of residues in disallowed regions. In contrast, I-TASSER 5.1 showed moderate accuracy, with more outliers than AlphaFold 3, while PEP-FOLD 4 consistently had the highest number of residues in disallowed regions, reflecting inferior structural predictions. This pattern held for other peptides, such as Liraglutide and Nesiritide, where AlphaFold 3 produced models with superior backbone geometry and minimized steric hindrance. The Ramachandran plots generated by I-TASSER 5.1, while moderately accurate, revealed slightly higher outliers, particularly in loop regions. In contrast, PEP-FOLD 4 models suffered from significant deviations, particularly in regions with high flexibility. Ramachandran plot validation further demonstrated AlphaFold 3’s superior structural accuracy. In a Ramachandran plot generated by PROCHECK, black dots represent the (phi, psi) dihedral angles of amino acid residues in a protein structure. These dots are distributed in regions based on the favorability of backbone dihedral angles. The yellow regions indicate the most favored conformations, while the red regions represent allowed conformations. Residues appearing as black dots in the yellow and red regions exhibit proper backbone geometry with minimal steric clashes, denoting a well-modeled structure. In contrast, black dots located outside these regions (disallowed regions) suggest structural issues, such as steric hindrance or improper backbone angles. The plot also uses symbols like squares and triangles to denote specific residue types, aiding in assessing residue-specific geometry. For all peptides, the models generated by AlphaFold 3 exhibited the fewest black dots in the disallowed regions, with the majority in the yellow and red regions, affirming the tool’s ability to produce high-quality structural predictions.

AlphaFold 3 also excelled in overall and local model quality metrics. For peptides like ANP and Exenatide, AlphaFold 3’s predictions aligned more closely with experimental structures, displaying lower deviations in global and localized structural features. I-TASSER 5.1 provided moderate results, often approximating the structural integrity observed in AlphaFold 3 models. However, PEP-FOLD 4 consistently underperformed, with inaccuracies in local model quality, particularly in active site regions or areas involving intricate secondary structures. While I-TASSER 5.1 demonstrated acceptable performance for simpler peptides, its accuracy declined for more structurally complex or larger peptides, such as FX06 and Apelin, where it struggled to maintain global structural fidelity. PEP-FOLD 4, optimized for generating de novo structures, was the least reliable, showing consistent issues with model quality, particularly regarding backbone geometry and residue orientations. In contrast, AlphaFold 3 consistently delivered high-quality results across peptides of varying complexity, reaffirming its utility in structural biology and peptide-based therapeutic research.

### 2.2. Evaluation of Molecular Docking Tools

Molecular docking is an essential computational tool for exploring peptide–receptor interactions, particularly in CAD research [21]. The docking analysis results across four prominent platforms (HADDOCK 2.4, HPEPDOCK 2.0, ClusPro 2.0, and HawkDock 2.0) highlight significant variations in binding affinity scores (presented as negative binding energies, where lower values indicate stronger binding) (Figure 1). These docking tools were evaluated for their performance in simulating the interaction of therapeutic peptides with CAD-related receptors, and their comparative performance underscores differences in algorithmic approaches, accuracy, and usability. HawkDock 2.0 consistently provided the most favorable binding scores for most peptide–receptor complexes. For example, in the interaction of atrial natriuretic peptide (ANP) with the receptors NPY1R and VEGFR2, HawkDock achieved binding scores of −20.1 and −20.9 kcal/mol, respectively. These values indicate a strong binding affinity, suggesting that HawkDock’s advanced scoring functions and energy minimization steps accurately simulate peptide–receptor interactions. The molecular mechanics generalized born surface area (MM/GBSA) approach employed by HawkDock is particularly advantageous, as it factors in solvent effects and provides a more accurate representation of the binding free energy [22,23], making it a superior choice for CAD-related docking studies. The method’s robustness in handling the complex interactions between therapeutic peptides and CAD receptors not only predicts binding affinities but also facilitates identifying key interactions crucial for therapeutic efficacy. ClusPro 2.0 demonstrated moderate to strong binding predictions in many cases but fell short of HawkDock in key receptor interactions. For instance, ClusPro produced binding scores of −9.5 and −10.4 kcal/mol for the ANP_NPY1R and Apelin_VEGFR2 complexes, noticeably weaker than HawkDock’s results. Despite its lower performance in some instances, ClusPro excelled in specific peptide–receptor pairs, such as Apelin_AT1R (−15.6 kcal/mol) and Exenatide_VEGFR2 (−12.9 kcal/mol). These successes highlight its utility for particular interactions where scoring functions align well with experimental findings. However, for broader applications across different receptor targets, ClusPro’s performance is less consistent compared to HawkDock, making it a supplementary tool rather than the primary choice for CAD-related docking studies.

HADDOCK 2.4 was competitive for a limited number of interactions, particularly in peptides targeting the interleukin-6 receptor (IL-6R). For example, HADDOCK provided a binding score of −12.1 kcal/mol for Exenatide_IL-6R, closely aligning with HawkDock (−12.9 kcal/mol). This performance demonstrates its capability to handle challenging peptide–receptor interactions with reasonable accuracy. However, its performance was inconsistent across other receptor targets, often yielding weaker binding scores compared to the other tools. HADDOCK’s approach to integrating experimental restraints through HADDOCK’s scoring system can be advantageous for specific applications. Still, its variable performance across different peptide–receptor combinations limits its overall utility for CAD docking studies. HPEPDOCK 2.0 generally delivered the weakest binding scores among the four platforms. While it showed some alignment with other tools for specific complexes, such as GLP-1_PDGFR (−10.0 kcal/mol), its overall reliability in predicting strong peptide–receptor binding was lower. For example, HPEPDOCK underperformed in Apelin_AT1R (−6.7 kcal/mol) and Exenatide_NPY1R (−10.6 kcal/mol) compared to HawkDock. The tool’s more straightforward scoring functions and less sophisticated energy models may contribute to its lower predictive accuracy. It appears less equipped to handle the intricacies of CAD-related interactions, making it less suitable for detailed peptide docking studies focused on CAD.

HawkDock 2.0 emerges as the most favorable platform for performing molecular docking of therapeutic peptides with CAD-related receptors. This superiority is attributed to its advanced MM/GBSA methodology, which provides a robust and detailed binding free energy calculation. HawkDock’s consistently strong performance across diverse peptide–receptor complexes, including challenging targets like NPY1R and VEGFR2, makes it a reliable choice for CAD-focused docking studies. The platform’s ability to accurately model solvent effects and predict binding affinities ensures that it captures the strength of interactions and the quality of the binding site complementarity, making it an ideal tool for CAD therapeutic development. Furthermore, HawkDock’s capability to handle a wide range of peptide–receptor interactions with its advanced scoring functions allows for better predicting biologically relevant binding affinities. Its integration of advanced computational techniques ensures high sensitivity and specificity in predicting the binding patterns of therapeutic peptides. This makes HawkDock particularly valuable for drug discovery and therapeutic peptide design in CAD, where precise prediction of binding interactions is crucial for developing effective treatments. The tool’s reliable performance across various CAD-related receptor targets provides a significant advantage in the preclinical evaluation of potential therapeutic peptides, enhancing the efficiency of the drug development pipeline.

Table 2 highlights the best binding affinity values for peptides functioning as antagonists. Among the peptides tested, Apelin demonstrated consistently strong binding to multiple receptors, including AT1R (−15.6 kcal/mol), β1AR (−17.8 kcal/mol), and IL-6R (−19.1 kcal/mol), as evaluated using the ClusPro 2.0 platform. Notably, Apelin also exhibited a high affinity for VEGFR2 (−21.7 kcal/mol) using HawkDock 2.0, the strongest binding affinity observed in this study. These findings suggest that Apelin could act as a potent antagonist against VEGFR2, a receptor integral to angiogenesis and vascular proliferation, highlighting its potential role in anti-angiogenic therapies. Additionally, ANP displayed a remarkable affinity for NPY1R (−20.1 kcal/mol) using HawkDock 2.0, demonstrating its potential as a potent antagonist for receptors associated with metabolic and cardiovascular regulation. On the other hand, Liraglutide exhibited moderate binding to MR (−11.0 kcal/mol) on the same platform, indicating a more selective antagonistic role.

Table 3 presents the best binding affinities for peptides functioning as agonists. Apelin emerged as a strong candidate, demonstrating a notable affinity for GLP-1R (−13.4 kcal/mol) on HawkDock 2.0 and APJ (−12.3 kcal/mol) on ClusPro 2.0. The APJ receptor is well-established for its role in cardiovascular homeostasis, further supporting Apelin’s relevance in CAD therapy. Exenatide exhibited high binding affinity to CaSR (−13.2 kcal/mol) using HADDOCK 2.4, suggesting its potential role in modulating calcium signaling for bone health and other metabolic processes. For LDLR, ANP showed a binding affinity of −11.0 kcal/mol on HPEPDOCK 2.0, reinforcing its application in lipid metabolism regulation. Furthermore, ANP displayed strong agonistic potential with A2A (−12.7 kcal/mol) using HawkDock 2.0, suggesting applications in neurovascular therapies. However, Apelin showed the most favorable profile among all the peptides studied due to its robust affinity for multiple cardiovascular-related receptors.

Apelin’s consistent and high-affinity interactions with key receptors, such as β1AR, AT1R, and APJ, highlight its unique potential in managing coronary artery disease (CAD). These receptors are central to cardiovascular regulation, including blood pressure control, vascular remodeling, and myocardial function. Apelin’s dual role as an antagonist (e.g., AT1R and VEGFR2) and agonist (e.g., APJ) further enhances its therapeutic flexibility, allowing for the modulation of both vasoconstrictive and vasodilatory pathways. In particular, the strong binding affinity of Apelin to VEGFR2 (−21.7 kcal/mol) positions it as a promising agent in anti-angiogenic therapies, which are crucial for reducing excessive vascular proliferation in CAD. Similarly, its interactions with AT1R and β1AR suggest its potential to mitigate hypertension and improve cardiac output, which is critical for CAD management. The complete molecular docking simulation results and molecular interactions can be seen in Appendix A, respectively.

### 2.3. Molecular Dynamics (MD) Simulation

The results from the MD simulations provide a detailed view of the interactions between therapeutic peptides and their respective target receptors (Table 4). These simulations were crucial in assessing the structural stability and binding affinities of different therapeutic peptides, focusing on Apelin as the most favorable candidate for CAD. Apelin consistently shows the lowest RMSD values across all its complexes with target receptors, ranging from 2.011 to 2.232 Å. These lower RMSD values indicate that Apelin binds more stably to its receptors, experiencing minimal structural deviations during the simulation period. For example, the Apelin_AT1R complex has an RMSD of 2.124 Å, suggesting a stable binding conformation throughout the simulation. In contrast, Liraglutide exhibits a higher RMSD (2.987 Å), indicating more significant conformational changes, which might compromise its binding efficiency and suitability for CAD treatment. The RMSF values offer insights into the flexibility of residues within the binding interface. Apelin complexes display lower RMSF values (1.511 to 1.867 Å), reflecting a more rigid and stable binding environment. For instance, Apelin_VEGFR2 shows the lowest RMSF of 1.511 Å, highlighting its optimal binding flexibility with minimal fluctuations. In comparison, complexes involving ANP and Exenatide demonstrate higher RMSF values (up to 2.708 Å), indicating more significant fluctuations and potentially weaker interactions with their respective receptors.

RoG values provide information about the compactness of the peptide–receptor complex. Apelin consistently presents a lower RoG (ranging from 2.326 to 2.519 Å) across its complexes, which implies a more compact and stable structure. The lower RoG indicates that Apelin maintains a stable conformation with minimal expansion upon binding. On the other hand, complexes involving FX06 and Exenatide show higher RoG values (up to 2.691 Å), which could imply less stability and more solvent accessibility in the binding pocket. Apelin complexes also exhibit a higher number of hydrogen bonds between the peptide and its receptor, ranging from 8 to 11 hydrogen bonds. This increased number of hydrogen bonds significantly contributes to the binding stability of Apelin. For example, Apelin_VEGFR2 forms 11 hydrogen bonds, which enhance its binding affinity. In contrast, complexes such as ANP_S1PR1 and Exenatide_CaSR have fewer hydrogen bonds (4–6), leading to less stable interactions and possibly weaker binding affinities. Apelin is the most favorable candidate for CAD treatment among all therapeutic peptides studied. Its consistently low RMSD, RMSF, and RoG values and a higher number of hydrogen bonds underscore its superior binding stability and affinity across different target receptors. Apelin’s binding conformation appears more favorable, maintaining a stable and compact structure essential for effective therapeutic action in CAD. The significant binding affinity of Apelin across various receptors like AT1R, β1AR, IL-6R, and VEGFR2, as evidenced by its lowest ΔG_binding_ values in both MM/PBSA calculations and MD simulations, further cements its potential as a leading therapeutic peptide for CAD.

### 2.4. Molecular Mechanics/Poisson–Boltzmann Surface Area (MM/PBSA) Calculations

The MM/PBSA calculations provided quantitative insights into therapeutic peptides’ binding free energy (ΔG_binding_) with their target proteins. More negative ΔG_binding_ values reflect the stronger binding and excellent thermodynamic stability of the peptide–protein complexes. Among the tested therapeutic peptides, Apelin displayed the most favorable binding affinity values, with ΔG_binding_ ranging from −81.69 to −199.17 kcal/mol (Table 5). Notably, its interaction with VEGFR2 yielded the most negative ΔG_binding_ (−199.17 kcal/mol), indicating an exceptionally stable complex. VEGFR2 plays a central role in angiogenesis, and inhibiting its activity can reduce excessive vascular proliferation, which is critical in the pathogenesis of CAD. Apelin also showed robust binding to AT1R (−140.54 kcal/mol) and β1AR (−156.53 kcal/mol), two key receptors in blood pressure regulation and myocardial function. These interactions suggest Apelin’s ability to modulate vasoconstriction and improve cardiac output, pivotal in CAD management. Furthermore, its binding to APJ (−171.62 kcal/mol) highlights its role in supporting cardiovascular homeostasis through vasodilatory and cardioprotective pathways. The ΔG_binding_ of Apelin with GLP-1R (−96.70 kcal/mol) and IL-6R (−163.66 kcal/mol) also indicates its potential to mitigate inflammation and improve glucose metabolism, further reinforcing its utility in addressing comorbidities often associated with CAD.

Binding free energy (ΔG_binding_) is a crucial parameter in evaluating the stability and affinity of therapeutic peptides when interacting with their target receptors. In the context of CAD therapy, lower binding free energy values indicate stronger and more stable interactions, which are often correlated with higher therapeutic potential. These interactions are essential for modulating critical biological pathways, such as reducing inflammation, improving endothelial function, and promoting vascular remodeling. For instance, peptides with significantly negative ΔG_binding_ values, such as Apelin with VEGFR2 (−199.17 kcal/mol) and APJ (−171.62 kcal/mol), suggest high binding stability, potentially enhancing their effectiveness in CAD treatment. However, while binding free energy provides insight into interaction strength, it must be complemented by experimental validation to confirm therapeutic efficacy and safety. Moreover, the receptor’s biological relevance to CAD pathology must be considered to ensure clinical applicability.

While other therapeutic peptides, such as ANP and Liraglutide, demonstrated moderate binding affinities, their ΔG_binding_ values were significantly less favorable compared to Apelin. For example, ANP exhibited a binding energy of −80.38 kcal/mol with NPY1R and −87.74 kcal/mol with A2A, suggesting potential in regulating metabolism and vascular tone. Similarly, Liraglutide showed a weak binding affinity of −50.27 kcal/mol with MR, indicating limited scope in CAD treatment. The MM/PBSA calculations emphasize Apelin’s superior binding performance across receptors critical to cardiovascular health. Its strong interaction with VEGFR2 suggests a significant role in anti-angiogenic therapy, targeting vascular remodeling and preventing atherosclerotic plaque formation in CAD patients. Meanwhile, its binding to AT1R and β1AR points to potential antihypertensive effects, enhancing myocardial efficiency and reducing the workload on the heart. The binding of Apelin to APJ and GLP-1R further underscores its versatility, as these receptors contribute to cardioprotection, glucose homeostasis, and inflammatory regulation, key factors in the systemic management of CAD.

## 3. Discussion

In therapeutic peptide design, integrating state-of-the-art computational tools like AlphaFold 3 and HawDock 2.0 offers a significant advantage. AlphaFold 3 is the best tool for the accurate 3D structure prediction of therapeutic peptides. Its deep-learning-based approach has revolutionized the field by providing high-fidelity models of peptide structures, facilitating a deeper understanding of their binding interactions and conformational flexibility. For therapeutic peptides targeting CAD, AlphaFold 3 can predict precise binding sites and conformations [24], which is critical for understanding their interaction with cardiovascular receptors like AT1R and β1AR. On the other hand, HawDock 2.0 excels in molecular docking simulations, providing robust performance in predicting peptide–receptor binding affinities and interactions. Its integration with MM/GBSA calculations enhances the accuracy of binding free energy predictions, which is crucial for evaluating the therapeutic potential of peptides for CAD. The advanced docking capabilities of HawDock 2.0 enable detailed analysis of the interaction dynamics between peptides and their target receptors, offering valuable insights into the stability and affinity of these complexes [22,25]. This combination of AlphaFold 3 for structure prediction and HawDock 2.0 for docking simulations significantly improves the reliability and applicability of computational methods in peptide-based drug discovery for CAD, paving the way for more precise therapeutic developments.

The molecular docking platform evaluation results highlight Apelin as the most favorable therapeutic peptide across different docking platforms. With binding affinities ranging from −15.6 to −21.7 kcal/mol across various target receptors, Apelin consistently demonstrates superior binding strength compared to other therapeutic peptides. The binding affinities for Apelin with AT1R, β1AR, IL-6R, PDGFR, ATP6AP2, and VEGFR2 underscore its potential as a strong candidate for CAD therapy. These findings align with previous research, where Apelin has been shown to exhibit significant vasodilatory effects and anti-inflammatory properties, which are critical in managing cardiovascular health [26,27,28]. Additionally, Apelin’s high binding affinity indicates its strong interaction potential with its target receptors, suggesting it could modulate these pathways effectively to manage CAD. This is consistent with experimental data showing Apelin’s ability to enhance endothelial function and reduce hypertension [29]. The alignment of Apelin’s pharmacological action with its binding affinities for these receptors implies that it could be beneficial in preventing and treating cardiovascular diseases. For instance, Apelin binding to receptors such as AT1R (angiotensin II receptor type 1) and β1AR (beta-1 adrenergic receptor) facilitates vasodilation [30], which enhances blood flow and reduces blood pressure, a key factor in preventing and managing CAD. MD simulations offer valuable insights into the dynamic properties of therapeutic peptide–receptor complexes. The RMSD, RMSF, RoG, and hydrogen bonding data reveal these complexes’ conformational flexibility and stability during simulation. Apelin consistently exhibits the lowest RMSD values (2.124 to 2.356 Å) across all complexes, indicating a stable and well-maintained conformation throughout the simulation. This stability is crucial for drug efficacy, suggesting that Apelin can maintain its binding structure without significant deviations over time [31,32]. The lower RMSF values observed for Apelin (1.511 to 1.867 Å) also indicate less fluctuation within its binding site than other peptides, indicative of a more rigid binding configuration and better stability. These results are consistent with findings from molecular dynamics studies of peptide–receptor interactions, where peptides with lower RMSF are considered to form more stable complexes, thereby enhancing their therapeutic potential [33,34]. Moreover, the higher number of hydrogen bonds Apelin forms with its receptors suggests more robust and stable interactions. Hydrogen bonds play a significant role in maintaining the structural integrity of protein–ligand complexes, and a higher number of such bonds enhances the binding affinity and stability of the therapeutic complex [35,36].

Comparing Apelin with other therapeutic peptides such as Liraglutide, ANP, and Exenatide, it becomes evident that Apelin shows superior binding affinity and maintains more excellent stability in its complexes. For instance, Liraglutide and Exenatide, while effective for diabetes [37,38], exhibit lower binding energies and weaker receptor interactions in the context of CAD. This comparison highlights the specific therapeutic advantages of Apelin in managing CAD. Previous studies have shown that Apelin modulates multiple signaling pathways, such as the renin–angiotensin system (RAS) and insulin signaling pathways [39,40], which are crucial for cardiovascular health. Its dual action on these pathways likely contributes to its enhanced therapeutic profile compared to other peptides, which typically target only one signaling pathway. For instance, Apelin has been shown to upregulate endothelial nitric oxide synthase (eNOS), leading to vasodilation and improved blood flow, which is beneficial in managing hypertension and heart failure [41,42].

### 3.1. Limitations, Clinical Implications, and Future Works

#### 3.1.1. Limitations

While providing valuable insights into the therapeutic potential of Apelin and other peptides for CAD, this study has its limitations. One primary limitation is the reliance on computational models for evaluating peptide–receptor interactions. Molecular docking and molecular dynamics simulations provide a detailed view of binding affinities and stability. Still, they do not account for the complex biological environment and the dynamic cellular conditions where these peptides would function in vivo. In particular, the study assumes static conformations of receptors and peptides during simulations, which may not accurately represent their behavior in a cellular milieu where conformational flexibility and dynamic interactions are crucial for therapeutic efficacy. Furthermore, the MM/PBSA calculations, although useful for estimating binding free energies, do not include solvation effects that could influence binding affinities in physiological conditions. Experimental validations such as surface plasmon resonance (SPR), bio-layer interferometry (BLI), or other biochemical assays are necessary to corroborate these computational findings. These methods would provide a more comprehensive understanding of the binding kinetics and thermodynamics, thereby validating the computational predictions made in this study. Another limitation is the study’s exclusion of in vivo models. While molecular dynamics simulations and docking studies offer valuable theoretical insights, they do not replicate the complex biological systems in animal models or humans. Thus, translating these findings into clinical applications requires subsequent in vivo studies to validate the therapeutic efficacy of Apelin and other peptides for CAD. In vivo experiments can help overcome the limitations of in vitro models, which often fail to recapitulate the physiological context, such as tissue-specific metabolism and cellular interactions within the cardiovascular system.

#### 3.1.2. Clinical Implications

Despite these limitations, the findings of this study have significant clinical implications for the management of CAD. With its superior binding affinities and dynamic stability, Apelin emerges as a promising therapeutic peptide for CAD, potentially offering a novel treatment strategy. The study’s identification of Apelin’s interaction with the multiple receptors involved in cardiovascular homeostasis positions it as a candidate for future clinical development. Its role in modulating the renin–angiotensin system, enhancing nitric oxide production, and reducing oxidative stress can provide therapeutic benefits beyond conventional treatments such as statins and ACE inhibitors [43,44]. Moreover, Apelin’s demonstrated efficacy in modulating inflammation and promoting vasodilation aligns well with the pathophysiology of CAD, providing a new avenue for treatment [8]. The results from this study suggest that Apelin could be beneficial in not only managing but also potentially preventing CAD in patients with risk factors such as hypertension and diabetes. The multifaceted role of Apelin in cardiovascular health may also help manage comorbid conditions, thereby improving overall cardiovascular outcomes. However, translating Apelin’s high binding affinities and stability from in silico findings to clinical applications presents several challenges. First, while in silico predictions suggest promising receptor binding, the actual pharmacokinetics and biodistribution of Apelin in vivo could differ significantly. Peptides often face challenges related to stability, bioavailability, and rapid degradation in the body, which may hinder their effectiveness when administered clinically. Furthermore, potential immunogenicity, which may arise from introducing a peptide into the body, could lead to adverse reactions or reduced therapeutic efficacy over time. Additionally, variability in patient receptor profiles and the presence of other comorbidities might affect how Apelin interacts with its targets, requiring further investigation into personalized therapeutic approaches. Finally, Apelin-based therapies’ long-term safety and efficacy must be evaluated through clinical trials before widespread use in CAD treatment. Additionally, the study’s findings could facilitate the design of personalized therapies. Apelin’s high binding affinity and stability profiles suggest that patients with specific receptor profiles could benefit most from Apelin-based therapies. Developing predictive biomarkers to identify such patients could lead to more targeted and individualized treatment regimens, maximizing therapeutic outcomes while minimizing side effects [45].

#### 3.1.3. Future Works

To build on the findings of this study, several avenues for future research can be pursued. First, experimental validation is crucial to confirm this study’s binding affinities and dynamics. In vitro and in vivo assays, such as SPR, BLI, and animal models, are needed to test Apelin’s efficacy and safety in a biological context. Such experiments would not only validate the computational predictions but also provide insights into the pharmacokinetics and pharmacodynamics of Apelin in vivo. Second, there is a need for further structural modifications of Apelin to improve its stability, bioavailability, and receptor specificity. Peptide engineering strategies such as cyclization, lipidation, or PEGylation could enhance Apelin’s therapeutic profile. Developing a stable, bioavailable form of Apelin that can be effectively administered would be a key step towards clinical application. Additionally, exploring combination therapies where Apelin is used alongside existing CAD treatments could provide synergistic benefits. For example, combining Apelin with anti-inflammatory drugs, anticoagulants, or statins could enhance therapeutic outcomes and broaden its clinical utility. Furthermore, understanding the molecular mechanisms by which Apelin interacts with different cardiovascular receptors could lead to the discovery of new therapeutic targets. Investigating Apelin’s downstream signaling pathways in detail would be essential for fully exploiting its therapeutic potential. Molecular and cellular studies to elucidate Apelin’s impact on endothelial function, vascular tone, and inflammation are warranted to understand its role in cardiovascular health comprehensively. Lastly, integrating artificial intelligence (AI) and machine learning (ML) into peptide-based drug discovery could accelerate the development of Apelin and other therapeutic peptides. Machine-learning models can predict peptide–receptor interactions with high accuracy, which could complement experimental and computational studies. AI can also aid in identifying potential biomarkers that predict patient response to Apelin therapy, thus enabling a more personalized approach to CAD treatment [46,47,48].

## 4. Materials and Methods

This study utilized a range of materials, including therapeutic peptides, receptor structures, and computational tools, to facilitate a comprehensive evaluation of peptide structure prediction and molecular docking and MD methodologies.

### 4.1. Selection of Therapeutic Peptides

The study focused on seven therapeutic peptides known for their clinical relevance to CAD, specifically those approved for clinical use or in advanced clinical trials. These peptides included FX06, Liraglutide, Exenatide, Nesiritide, Apelin, GLP-1, and Atrial Natriuretic Peptide (ANP). The selection criteria prioritized peptides with documented roles in modulating key pathophysiological mechanisms associated with CAD, such as vascular inflammation, endothelial dysfunction, cardiac remodeling, and blood pressure regulation. This ensured that the peptides studied had a direct and meaningful connection to CAD pathophysiology. A systematic approach was adopted to identify peptides for docking studies. First, clinical literature and database searches were conducted to compile a list of candidate peptides with known or investigational relevance to CAD. Emphasis was placed on peptides targeting critical pathways, including renin–angiotensin–aldosterone signaling, nitric oxide bioavailability, and inflammatory cytokine modulation. Next, peptides with demonstrated efficacy or potential in mitigating CAD-related complications were shortlisted, ensuring diverse mechanisms of action, such as receptor agonism, antagonism, or enzyme inhibition. The amino acid sequences of the selected peptides were retrieved from reliable public databases, including UniProt version 2024, UniProt Consortium, Hinxton, UK [49] and the Therapeutic Target Database (TTD) version 2024, nnovative Drug Research and Bioinformatics Group, Zhejiang University, Hangzhou, China [50], ensuring the input data’s integrity, accuracy, and reliability for structural modeling and subsequent computational analyses. Before proceeding with structure prediction, the retrieved sequences were verified for completeness and correctness.

These peptides served as a foundation for evaluating the performance of peptide structure prediction and docking tools in this research. The docking studies focused on key receptor interactions relevant to CAD, such as the APJ receptor for Apelin, β-adrenergic receptors for ANP, and GLP-1 receptors for Exenatide and GLP-1. The study aimed to bridge the gap between computational modeling and translational research by selecting peptides with established or investigational roles in CAD pathways. The selection also incorporated therapeutic diversity to evaluate the computational tools across varying structural and functional peptide classes. For example, FX06 is known for its anti-inflammatory effects, Liraglutide and Exenatide target glucose regulation, and cardiometabolic health, while Nesiritide and ANP are natriuretic peptides involved in vasodilation and fluid balance. Table 6 summarizes the therapeutic peptides used in this study, their amino acid sequences, and the binding sites identified for receptor interactions.

### 4.2. Target Receptor Structures

The receptors selected for the docking studies were carefully chosen based on their direct involvement in the pathophysiology of CAD. These receptors play pivotal roles in vascular remodeling, inflammation, blood pressure regulation, and other critical processes that contribute to the progression of CAD. Among the selected receptors were the Angiotensin II type 1 receptor (AT1R), Glucagon-like peptide-1 receptor (GLP-1R), Interleukin-6 receptor (IL-6R), and several other receptors implicated in smooth muscle proliferation, endothelial function, and inflammation (Table 7). To ensure accurate docking results, high-resolution protein structures of these receptors were retrieved from the Protein Data Bank (PDB) version 2024, Research Collaboratory for Structural Bioinformatics (RCSB), Piscataway, NJ, USA, prioritizing structures obtained via X-ray crystallography or cryo-electron microscopy (cryo-EM). In the context of CAD, some of the receptors selected function as antagonists, aiming to block detrimental pathways (such as vasoconstriction and inflammation). In contrast, others are agonists, promoting beneficial effects such as vasodilation and improved cardiac function. These receptor structures provided the foundation for assessing the effectiveness of therapeutic peptides targeting CAD in this study. The binding sites of these selected receptors were further analyzed using CASTp version 3.0, Jie Liang’s Lab, University of Illinois at Chicago, Chicago, IL, USA [51], a tool that provided insights into the structural features and potential interaction sites, facilitating a more precise assessment of therapeutic peptide efficacy in targeting CAD-related pathways. The complete therapeutic peptides and receptors dataset can be seen in Appendix A.

### 4.3. Selection of Computational Tools

Three advanced, web-based tools were utilized to predict the 3D structures of the selected peptides: AlphaFold version 3, DeepMind, London, UK; I-TASSER 5.1 version 5.1, Zhang Lab, University of Michigan, Ann Arbor, MI, USA; and PEP-FOLD version 4, Structural Bioinformatics Unit, French National Institute for Agricultural Research (INRA), Paris, France. These tools were chosen based on their complementary methodologies, proven accuracy in peptide structure prediction, and ease of use, ensuring a robust analysis across varying peptide characteristics. Each tool offered unique approaches suited to different peptide characteristics, ensuring accurate and reliable structure predictions. AlphaFold 3 leverages cutting-edge deep-learning algorithms to predict highly accurate peptide and protein structures [24]. By analyzing sequence data and applying evolutionary insights, AlphaFold 3 generates predictions with remarkable precision, even without experimental structural data. One of its key strengths lies in its ability to effectively model both small and large peptides, making it an essential resource for studying therapeutic peptides. However, its performance can be less reliable for highly disordered regions or sequences lacking evolutionary conservation, sometimes in therapeutic peptides. This tool is particularly valuable for predicting peptide conformations, as it can model both small and large peptides with a high degree of accuracy, making it an essential resource in structural biology [66]. I-TASSER 5.1 combines template-based modeling with ab initio methods, providing robust peptide predictions, especially when experimental data is limited. This hybrid approach allows I-TASSER 5.1 to build reliable models by identifying homologous templates from existing databases and using them to predict the peptide’s 3D structure. For sequences that are less homologous to known structures, I-TASSER 5.1 employs ab initio strategies to explore alternative folding pathways, ensuring comprehensive structural coverage and enhancing its reliability for peptide prediction [67,68]. While this increases its reliability, the accuracy of predictions can decrease for peptides with highly flexible or unique conformations, as the tool relies partially on the availability of structural templates. Nevertheless, its integration of multiple methodologies makes it a versatile option for diverse peptide studies. PEP-FOLD 4, specifically designed for short peptides, utilizes a fragment-based approach that assembles small segments of known peptide structures to generate potential conformations [69]. This method provides accurate 3D models by focusing on the local backbone geometry, which is crucial when working with therapeutic peptides that typically consist of fewer than 50 amino acids. PEP-FOLD 4 is particularly valuable for peptides involved in receptor binding and drug design, as it ensures reliable conformational predictions for short peptides [70,71]. However, its reliance on predefined fragment libraries may limit its utility for longer or highly disordered peptides. Despite this limitation, PEP-FOLD 4 is particularly suited for therapeutic peptides, where receptor-binding conformations are critical.

For molecular docking, four well-established web-based platforms were selected: HADDOCK version 2.4, Computational Structural Biology Group, Utrecht University, Utrecht, The Netherlands; HPEPDOCK version 2.0, the College of Life Sciences, Nankai University, Tianjin, China; ClusPro version 2.0, the Structural Bioinformatics Lab, Boston University, Boston, MA, USA; and HawDock version 2.0, he Department of Physics, Huazhong University of Science and Technology, Wuhan, China. These platforms were chosen based on their proven capabilities to model peptide–receptor interactions, each offering unique approaches to docking, ranging from rigid-body to flexible docking techniques. The primary reason for selecting these platforms is their web-based availability, which ensures accessibility without complex installations. Additionally, their ease of use simplifies the docking process, making them suitable for researchers at varying levels of expertise. Furthermore, these platforms provide advanced features such as specifying active binding sites, enhancing the precision of docking simulations, and enabling more reliable interaction models. HADDOCK 2.4 integrates experimental data such as mutagenesis or NMR data to guide the docking process, improving the accuracy of peptide–receptor interaction predictions. Using such data, HADDOCK can generate more reliable models that better reflect the binding dynamics of peptides to their target receptors [72]. HPEPDOCK 2.0 is optimized for docking peptides to large protein structures, offering a fast and efficient algorithm designed for high-throughput applications. This tool is particularly effective when working with large receptor structures, as it efficiently handles the complexity of peptide–protein interactions [73]. ClusPro 2.0 focuses on generating multiple binding poses, allowing for a more comprehensive analysis of peptide–receptor interactions. By considering the peptide’s and receptor’s flexibility, ClusPro provides a broader range of possible binding orientations, which helps identify the most stable and relevant binding conformations [74]. HawDock 2.0 stands out for its ability to leverage sequence-based approaches to improve docking precision, particularly for peptide–protein complexes. By integrating evolutionary information and structural features, HawDock delivers refined docking results, making it highly effective for studying complex interactions [75]. These docking platforms offer a range of strategies for evaluating peptide–receptor interactions, from rigid-body docking, where the peptide and receptor are treated as fixed entities, to flexible docking, which allows for conformational changes in both the peptide and the receptor during the docking process.

### 4.4. Computing Power

The MD simulations were conducted on a high-performance workstation with the following specifications: an Intel^®^ Core™ i9-12900KF CPU (3.90 GHz, 16 cores), Intel Corporation, Santa Clara, CA, USA; an NVIDIA GeForce RTX 4090 GPU with 24 GB GDDR6X memory, NVIDIA Corporation, Clara, CA, USA; 64 GB DDR5 RAM, Corsair, Fremont, CA, USA; an 8 TB HDD storage, Western Digital, San Jose, CA, USA; and running the Ubuntu operating system, Canonical Ltd., London, UK.

### 4.5. 3D Structure Modeling

The structure prediction pipeline employed in this study followed a systematic approach to ensure the generation of reliable peptide models for subsequent docking studies. The process involved several key steps, starting with sequence preparation and ending with the validation and comparison of the predicted structures.

Sequence Preparation: The first step in the structure modeling pipeline involved formatting the amino acid sequences of the therapeutic peptides according to the requirements of each structure prediction tool. The sequences were thoroughly checked for completeness and alignment with available experimental data.Prediction with AlphaFold 3: The next step in the pipeline involved using AlphaFold 3, a deep-learning-based tool that predicts high-resolution atomic structures. Peptide sequences were input into AlphaFold 3′s web interface, where the tool utilized deep neural networks to generate 3D structures. The accuracy of the predicted structures was assessed based on per-residue confidence scores (pLDDT), which indicate how reliable the predictions are.Modeling with I-TASSER 5.1: The sequences were also modeled using I-TASSER 5.1, a tool that combines template-based modeling with ab initio simulations for additional structural insights. Peptide sequences were aligned against a template library to identify structurally similar proteins. The tool’s threading algorithms generated initial models, which were then refined through iterative ab initio simulations. The quality of the resulting models was assessed using the C-score and TM-score, with higher values indicating better model quality.De Novo Prediction with PEP-FOLD 4: To complement the results from AlphaFold 3 and I-TASSER 5.1, PEP-FOLD 4 was used for de novo prediction of peptide structures. PEP-FOLD 4 generated conformational ensembles for each peptide using fragment libraries and performing energy minimization.Structural Optimization: All models underwent a structural optimization process once the initial structures were predicted. This involved energy minimization using molecular mechanics force fields such as AMBER or OPLS to remove steric clashes and improve the overall stability of the structures.Validation of Structures: Several validation tools were used to ensure the quality and reliability of the predicted structures. Stereochemical analysis was conducted using Ramachandran plots generated by PROCHECK version 3.5.4, Biocomputing Unit, Institute of Cancer Research, London, UK [76] to assess the backbone dihedral angles. Additionally, MolProbity version 4.5, the Richardson Lab, Duke University, Durham, NC, USA [77] was used to identify potential structural outliers and errors in the predicted models. Any structures showing significant deviations were refined using visualization and editing tools such as PyMOL version 2.5, Schrödinger, LLC, New York, NY, USA [78] and Chimera version 1.16, University of California, San Francisco (UCSF), San Francisco, CA, USA [79].Comparison Across Tools: Finally, the structures predicted by AlphaFold 3, I-TASSER 5.1, and PEP-FOLD 4 were compared to identify the most reliable confirmations. Key parameters such as secondary structure elements, folding patterns, and overall energy were analyzed. Based on these comparisons, the best model for each therapeutic peptide was selected for use in the subsequent molecular docking simulations.

### 4.6. Peptide–Protein Docking Simulation

Peptide–protein docking simulations are crucial for understanding the interaction between therapeutic peptides and their target receptors, which is an essential step in developing peptide-based therapeutics. These simulations help predict peptides’ binding mode, affinity, and specificity when interacting with proteins. Receptor structures were retrieved from the PDB, and missing residues were analyzed and refined using Swiss-PdbViewer version 4.1.0, the Swiss Institute of Bioinformatics (SIB), Lausanne, Switzerland [80]. The peptides were energy-minimized to resolve steric clashes and ensure stable conformations. The final optimized receptor and peptide structures were prepared for docking simulations to predict peptide–receptor interactions. Docking simulations were performed using a variety of web-based platforms, including HADDOCK 2.4, HPEPDOCK 2.0, ClusPro 2.0, and HawDock 2.0. These tools utilized flexible docking, allowing for conformational changes in the peptide and receptor during the docking process. The simulations generated various binding poses, each with an associated docking score that reflects the strength of the interaction between the peptide and the receptor. Once the docking simulations were completed, the resulting binding poses were ranked based on their docking scores. These scores were calculated by considering multiple factors, including hydrogen bonds, electrostatic interactions, van der Waals forces, and solvation energy. To ensure consistency in comparing docking scores across the different platforms, the PRODIGY (PROtein binDIng enerGY prediction), Bijvoet Centre for Biomolecular Research, Utrecht University, Utrecht, The Netherlands [81,82] tool was employed. PRODIGY standardizes docking score annotations, allowing for normalizing and directly comparing scores derived from various docking tools. This process facilitated the ranking of binding poses, where higher docking scores indicated stronger binding interactions. The top-ranked binding poses underwent rigorous validation involving visual inspection with molecular visualization tools. Key interaction details, such as hydrogen bonds, hydrophobic interactions, and critical binding residues, were thoroughly analyzed. Following this comprehensive evaluation, the most stable and energetically favorable peptide–receptor complexes were selected for subsequent MD simulations to assess their dynamic behavior and stability in a biologically relevant environment.

### 4.7. Molecular Dynamics (MD) Simulation

The MD simulations were employed to examine the dynamics and stability of protein–peptide complexes, specifically targeting therapeutic peptides in clinical use for CAD trials. The simulations were conducted using GROMACS version 2022.5, KTH Royal Institute of Technology and Uppsala University, Stockholm, Sweden [83], a widely used software due to its precision in modeling biomolecular systems. The optimized potentials for liquid simulations (OPLS-AA/L) force field [84] was applied to accurately model the molecular interactions within the complexes. Simulation box dimensions were set using default cubic parameters to appropriately accommodate the biomolecular complexes. The system was prepared by adding water molecules with the Single Point Charge Extended (SPCE) model and incorporating counterions to ensure system neutrality [85]. Energy minimization was performed using the steepest-descent method to resolve steric clashes and stabilize the system. The equilibration process was executed in two phases: phase 1, in the NVT ensemble, regulated temperature fluctuations, while phase 2, in the NPT ensemble, maintained constant pressure and temperature [86]. After equilibration, production MD simulations were run over 100 nanoseconds to observe the protein–peptide complex dynamics. During the simulations, key parameters such as root mean square deviation (RMSD), root mean square fluctuation (RMSF), radius of gyration (RoG), potential energy, and intermolecular hydrogen bonding interactions were monitored to evaluate complex stability and conformational dynamics. Visualization and analysis of critical residues and intermolecular interactions were performed using PyMOL version 2.5, Schrödinger, LLC, New York, NY, USA and UCSF Chimera version 1.16, University of California, San Francisco (UCSF), San Francisco, CA, USA. These analyses provided insights into therapeutic peptides’ binding mechanisms and stability-targeting CAD receptors, offering valuable data for their clinical application and trials.

### 4.8. Molecular Mechanics/Poisson–Boltzmann Surface Area (MM/PBSA) Calculations

The Molecular Mechanics/Poisson–Boltzmann Surface Area (MM/PBSA) method, integrated with MD simulations, assessed peptide–protein interactions, focusing on therapeutic peptides targeting CAD. MD simulations provided a range of conformations, and representative snapshots were selected for detailed energy analysis. These snapshots underwent energy calculations, including gas-phase energy, solvation energy estimation using a continuum solvent model, and entropy computations [87]. The results were used to determine the binding free energy of the protein–peptide complex. For this analysis, the *gmx_MMPBSA* module within the GROMACS simulation package was employed to efficiently compute the binding free energies [88,89]. The MM/PBSA method is recognized for its accuracy in predicting the binding free energy of peptide–protein interactions, making it an essential tool for evaluating the energetics of biomolecular complexes [90]. The binding free energy was calculated using the following equation:ΔG_binding_ = ΔG_complex_ − ΔG_peptide_ − ΔG_protein_
where:ΔG_binding_: the binding free energy associated with forming the peptide–protein complex.ΔG_complex_: the free energy of the fully solvated peptide–protein complex.ΔG_peptide_: the free energy of the peptide in its solvated state when unbound.ΔG_protein_: the free energy of the protein in its solvated state when unbound.

This calculation provided valuable insights into the energetic changes during peptide–protein complex formation. It offered a deeper understanding of the stability and strength of these interactions, which is critical for evaluating the therapeutic potential of peptides in CAD treatment.

### 4.9. Statistical Analysis

Statistical analysis was conducted in this study to examine the relationships among the parameters obtained from molecular docking and molecular dynamics simulations. The data from these computational experiments were rigorously analyzed and interpreted using OriginLab Pro version 2022, OriginLab Corporation, Northampton, MA, USA [91]. This approach facilitated the identification of correlations, trends, and patterns within the dataset, providing meaningful insights into the interrelationships between the different variables and parameters.

## 5. Conclusions

In conclusion, this study underscores the potential of therapeutic peptides, particularly Apelin, in targeting cardiovascular disease through their interaction with specific receptors. Utilizing advanced computational methods such as AlphaFold 3 for 3D structure prediction and HawDock 2.0 for molecular docking simulations, the findings reveal Apelin’s exceptional binding affinities and stability, making it a promising candidate for CAD therapy. Apelin demonstrates superior binding strengths with key receptors involved in cardiovascular health, alongside its ability to modulate multiple signaling pathways, which are crucial for managing hypertension and heart failure. The study highlights the importance of these computational tools in enhancing our understanding of peptide–receptor interactions and the therapeutic potential of peptides. Future research should explore Apelin’s pharmacokinetics, pharmacodynamics, and potential combination therapies in clinical settings to fully realize its therapeutic benefits and refine personalized treatment strategies for CAD.

## Figures and Tables

**Figure 1 ijms-26-00462-f001:**
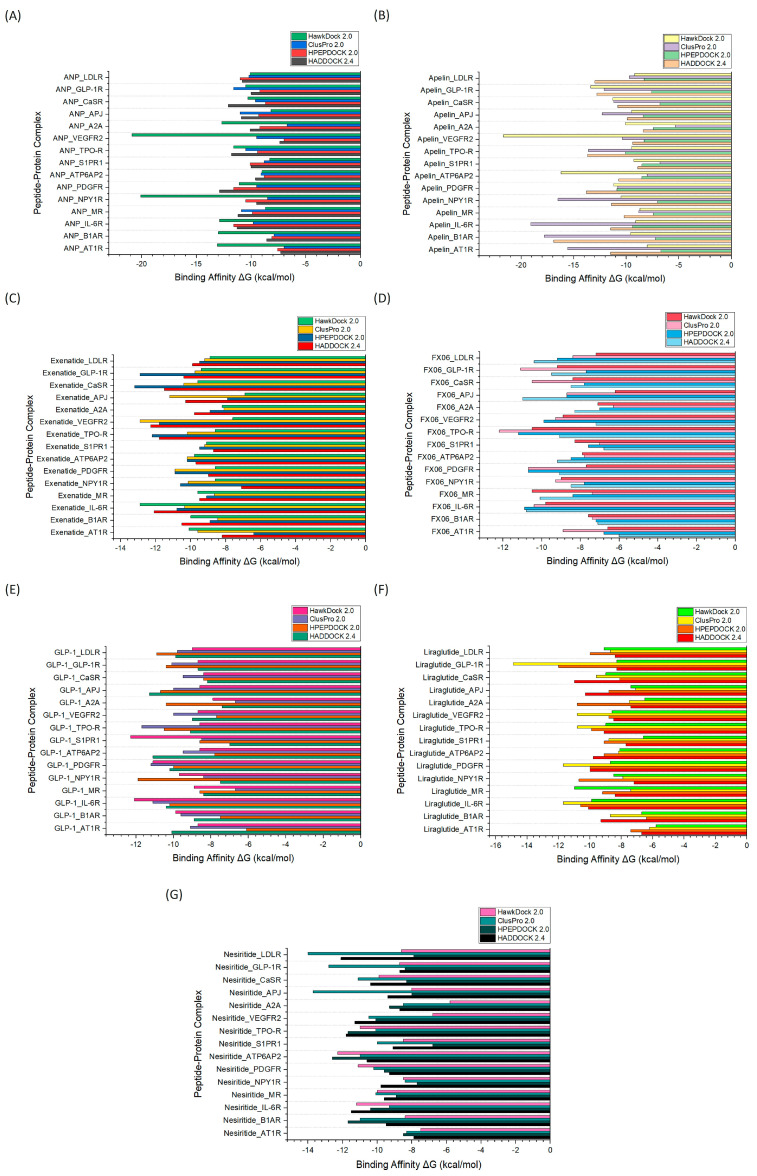
Comparison of molecular docking simulation results from HADDOCK 2.4, HPEPDOCK 2.0, ClusPro 2.0, and HawDock 2.0, showing the binding affinity (ΔG, kcal/mol) of seven therapeutic peptides interacting with CAD-related receptors. (**A**) ANP peptide-protein complexes. (**B**) Apelin peptide-protein complexes. (**C**) Exenatide peptide-protein complexes. (**D**) FX06 peptide-protein complexes. (**E**) GLP-1 peptide-protein complexes. (**F**) Liraglutide peptide-protein complexes. (**G**) Nesiritide peptide-protein complexes.

**Table 1 ijms-26-00462-t001:** Comparison of peptide (Apelin) structure prediction metrics across AlphaFold 3, I-TASSER 5.1, and PEP-FOLD 4.

Peptide	Z-Score	Overall Model Quality	Local Model Quality	Ramachandran Plot
Apelin				
AlphaFold 3	−4.21	* 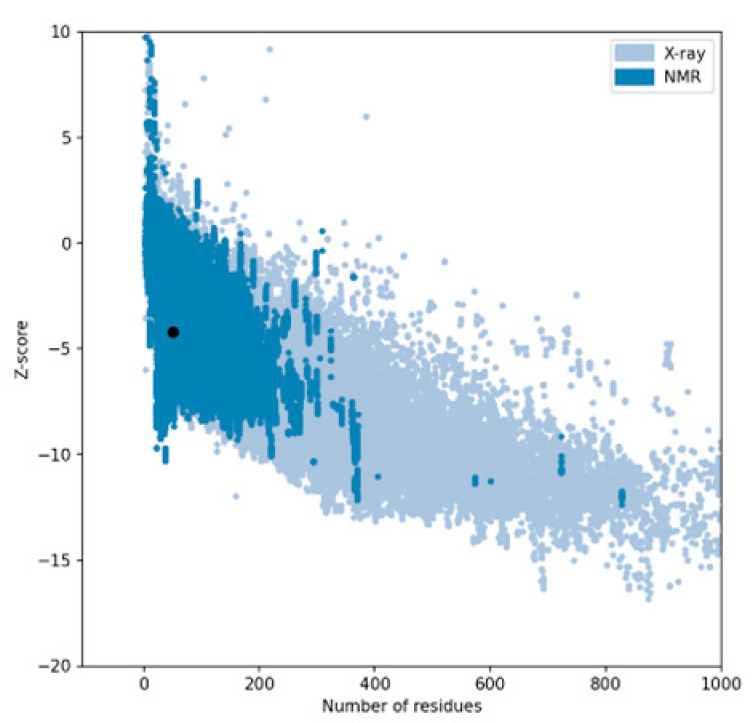 *	* 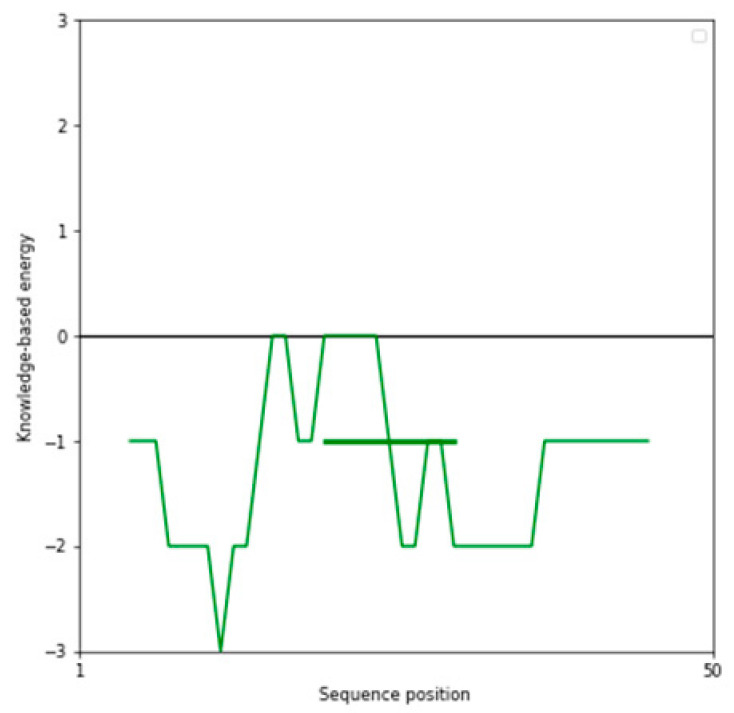 *	* 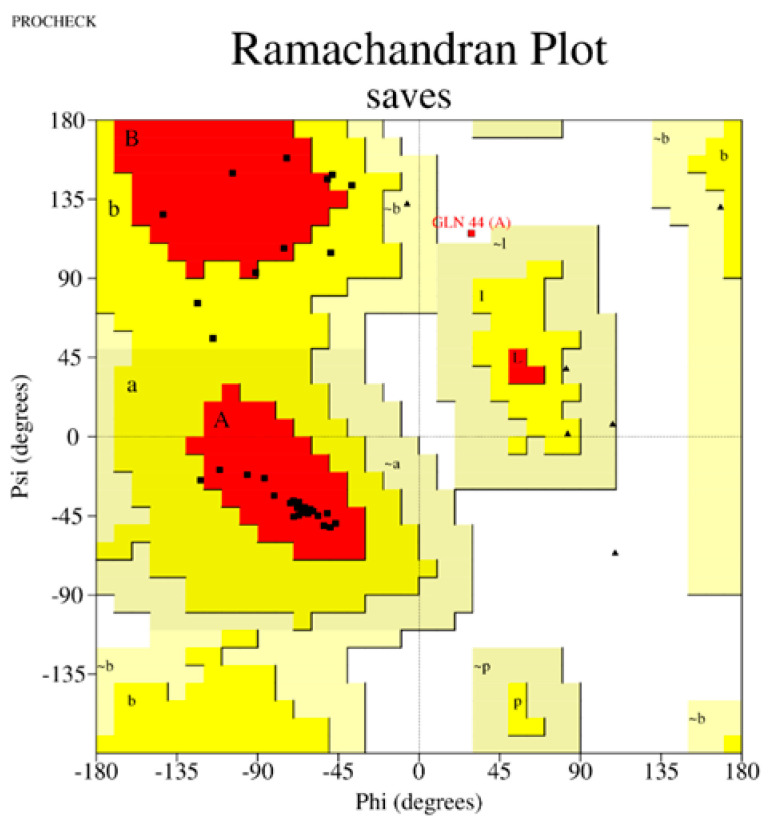 *
I-TASSER 5.1	−2.06	* 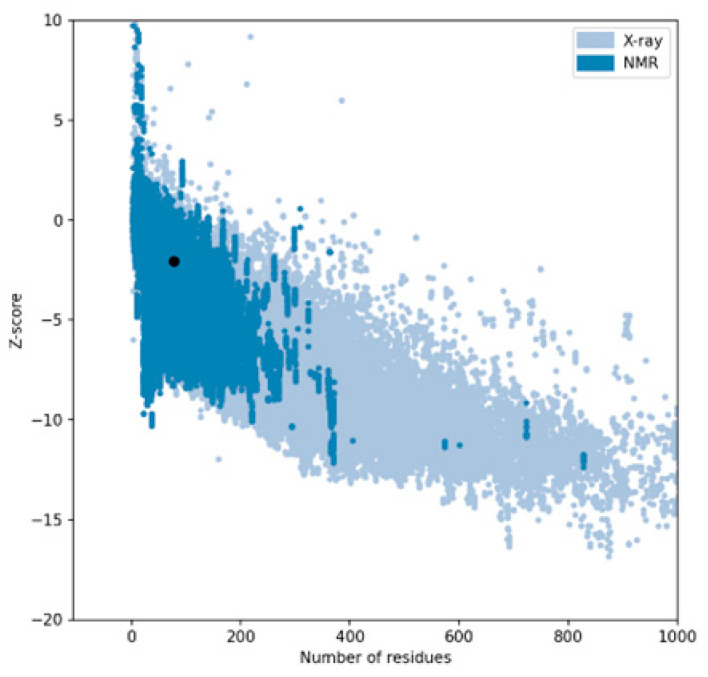 *	* 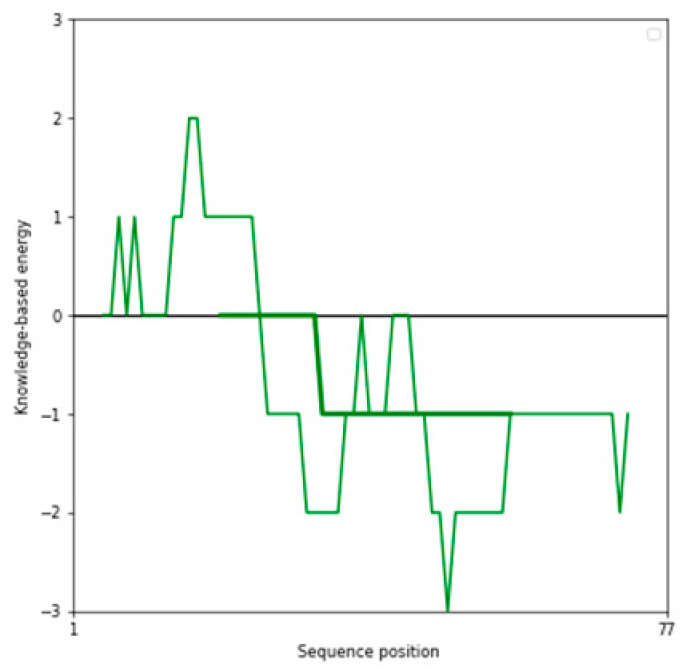 *	* 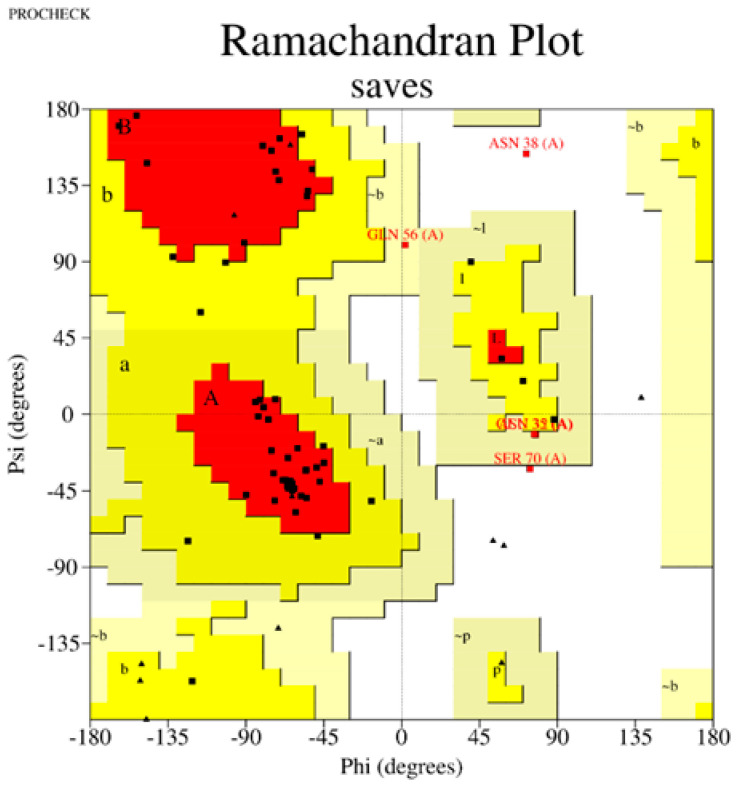 *
PEP-FOLD 4	−1.15	* 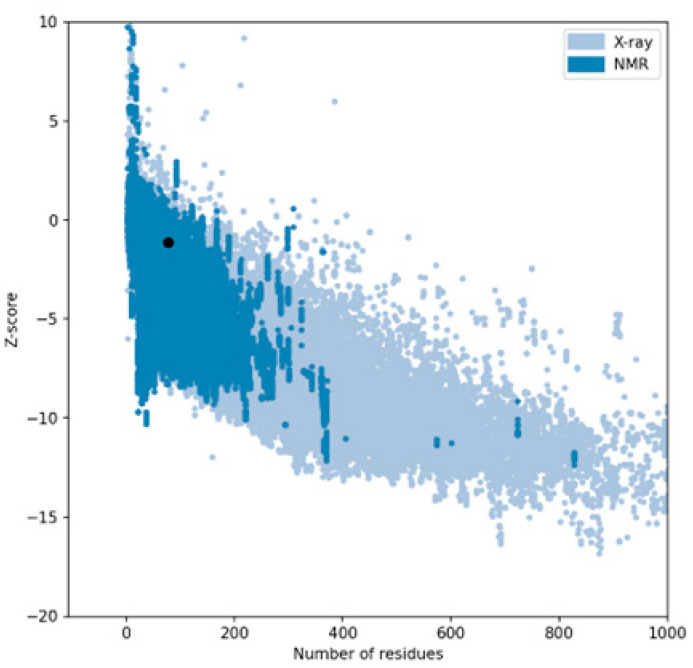 *	* 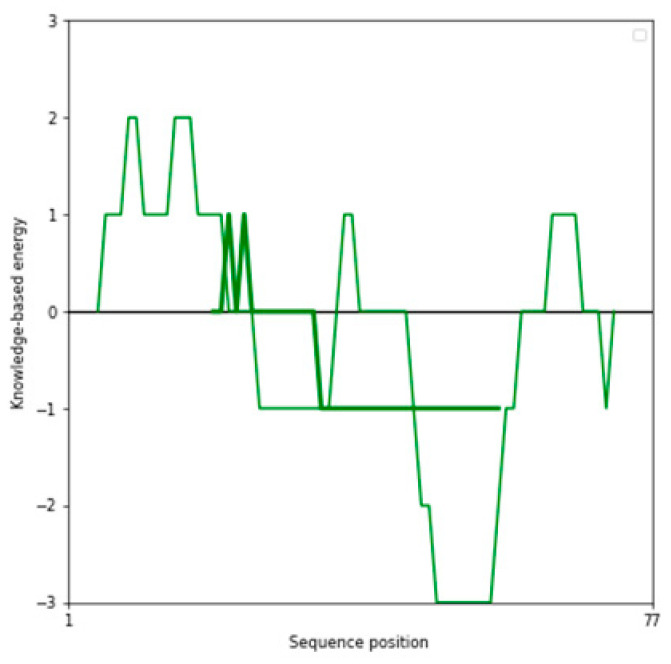 *	* 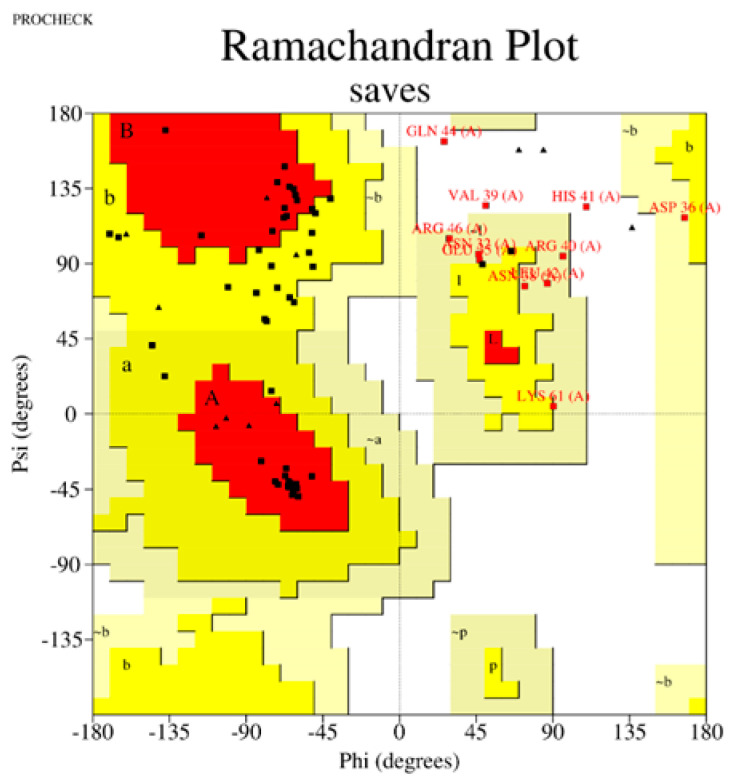 *

**Table 2 ijms-26-00462-t002:** The best binding affinity results of therapeutic peptides as antagonists against target receptors across docking platforms.

Receptor	Therapeutic Peptide	Docking Platform	Binding Affinity (kcal/mol)
AT1R	Apelin	ClusPro 2.0	−15.6
β1AR	Apelin	ClusPro 2.0	−17.8
IL-6R	Apelin	ClusPro 2.0	−19.1
MR	Liraglutide	HawkDock 2.0	−11.0
NPY1R	ANP	HawkDock 2.0	−20.1
PDGFR	Apelin	ClusPro 2.0	−13.8
ATP6AP2	Apelin	HawkDock 2.0	−16.2
S1PR1	ANP	HADDOCK 2.4	−10.0
TPO-R	FX06	ClusPro 2.0	−12.2
VEGFR2	Apelin	HawkDock 2.0	−21.7

**Table 3 ijms-26-00462-t003:** The best binding affinity results of therapeutic peptides as agonists against target receptors across docking platforms.

Receptor	Therapeutic Peptide	Docking Platform	Binding Affinity (kcal/mol)
A2A	ANP	HawkDock 2.0	−12.7
APJ	Apelin	ClusPro 2.0	−12.3
CaSR	Exenatide	HADDOCK 2.4	−13.2
GLP-1R	Apelin	HawkDock 2.0	−13.4
LDLR	ANP	HPEPDOCK 2.0	−11.0

**Table 4 ijms-26-00462-t004:** Molecular dynamics (MD) simulation results of the 15 best therapeutic peptide–protein complexes for CAD.

Therapeutic Peptide–Protein Complex	Average RMSD (Å)	Average RMSF (Å)	Average RoG (Å)	Number of Hydrogen Bonds Between the Two Proteins
Apelin_AT1R	2.124	1.821	2.352	8
Apelin_β1AR	2.267	1.744	2.342	10
Apelin_IL-6R	2.011	1.637	2.346	9
Liraglutide_MR	2.987	2.378	2.632	5
ANP_NPY1R	2.824	2.412	2.639	6
Apelin_PDGFR	2.357	2.004	2.499	7
Apelin_ATP6AP2	2.139	1.904	2.519	8
ANP_S1PR1	2.902	2.554	2.652	4
FX06_TPO-R	2.706	2.228	2.629	5
Apelin_VEGFR2	2.008	1.511	2.326	11
ANP_A2A	2.675	2.425	2.653	6
Apelin_APJ	2.232	1.867	2.446	10
Exenatide_CaSR	3.012	2.708	2.691	4
Apelin_GLP-1R	2.356	1.912	2.487	9
ANP_LDLR	2.899	2.509	2.624	5

**Table 5 ijms-26-00462-t005:** MM/PBSA calculation results for binding free energies of the best therapeutic peptide–protein complexes.

Therapeutic Peptide–Protein Complex	MM/PBSA Calculation Results ΔG_binding_ (kcal/mol)
Apelin_AT1R	−140.54
Apelin_β1AR	−156.53
Apelin_IL-6R	−163.66
Liraglutide_MR	−50.27
ANP_NPY1R	−80.38
Apelin_PDGFR	−81.69
Apelin_ATP6AP2	−103.43
ANP_S1PR1	−54.57
FX06_TPO-R	−60.90
Apelin_VEGFR2	−199.17
ANP_A2A	−87.74
Apelin_APJ	−171.62
Exenatide_CaSR	−49.35
Apelin_GLP-1R	−96.70
ANP_LDLR	−49.26

**Table 6 ijms-26-00462-t006:** The therapeutic peptides for CAD that are either in current use or undergoing clinical trials, as well as their sequences and key binding sites (residue positions).

Therapeutic Peptide	Sequence	Binding Site (Position of Residues)
ANP	SLRRSSCFGGRMDRIGAQSGLGCNSFRY	4, 5, 6, 7, 8, 10, 11, 12, 14, 15, 23, 24, 26
Apelin	MNLRLCVQALLLLWLSLTAVCGGSLMPLPD	10, 11, 14, 15, 25, 26, 27, 28, 29
Exenatide	HGEGTFTSDLSKQMEEEAVRLFIEWLKNGG	14, 17, 18, 21
FX06	MKHLLLLLLCVFLVKSQGVNDNEEGFFS	13, 17, 19, 24
GLP-1	HAEGTFTTSDVSYSSTLEGQAAKEFIAWLV	1, 3, 4, 5, 6, 10, 11, 14
Liraglutide	HAEGTFTSDVSSYLEGQAAKEEFIAWLVRG	1, 2, 3, 10, 13, 14
Nesiritide	SPKMVQGSGCFGRKMDRISSSSGLGCKVLR	4, 5, 6, 15, 19, 29, 32

**Table 7 ijms-26-00462-t007:** Structural information and binding site residues of receptors targeted in docking studies for therapeutic peptides in CAD pathophysiology.

Receptor	PDB ID	Binding Site (Position of Residues)	Note
**Therapeutic Peptide Role: Antagonist**			
Angiotensin II Type 1 Receptor (AT1R)	4YAY [52]	35, 84, 88, 105, 108, 109, 112, 163, 167, 182, 288, 292	To reduce vasoconstriction and blood pressure
Beta-adrenergic Receptor (β1AR)	7BTS [53]	884, 899, 940, 942, 943, 955, 966, 970, 1004, 1005, 1006, 1007, 1009, 1015, 1022, 1032, 1033, 1034, 1035, 1038, 1057, 1121, 1123, 1138, 1209, 1212, 1215, 1218, 1221, 1222, 1228, 1232, 1234, 1321, 1340, 1344, 1349, 1350, 1363, 1364, 1379, 1384	To reduce heart rate and workload on the hear
Interleukin-6 Receptor (IL-6R)	1N26 [54]	46, 69, 72, 90, 91, 92, 122, 123, 124	To reduce inflammation and immune response
Mineralocorticoid Receptor (MR)	1Y9R [55]	769, 770, 772, 773, 776, 807, 810, 814, 817, 845, 941, 942, 945, 954	To prevent sodium retention and reduce blood pressure
Neuropeptide Y Receptor Y1 (NPY1R)	7VGX [56]	117, 120, 121, 124, 173, 200, 212, 215, 219, 220, 280, 282, 283, 284, 286, 287, 294, 298, 299, 302	To reduce vasoconstriction and sympathetic nervous system activity
Platelet-Derived Growth Factor Receptor Alpha (PDGFR)	6JOL [57]	625, 627, 644, 648, 658, 674, 676, 677, 814, 815, 816, 825, 835, 836, 837	To inhibit smooth muscle proliferation and plaque formation
Renin Receptor (ATP6AP2)	3LC8 [58]	82, 155, 210, 230, 287, 306, 310, 363, 366, 367, 368, 370	To inhibit renin activity and reduce blood pressure
Sphingosine-1-Phosphate Receptor 1 (S1PR1)	7TD3 [59]	29, 34, 46, 53, 98, 110, 121, 175, 210, 276, 297	To inhibit smooth muscle proliferation and vascular inflammation
Thrombopoietin Receptor (TPO-R)	8G04 [60]	288, 290, 291, 292, 300, 302, 303, 304, 349, 390, 473, 475, 476, 477	To reduce platelet activation and prevent thrombosis
Vascular Endothelial Growth Factor Receptor 2 (VEGFR2)	4ASD [61]	840, 848, 866, 868, 885, 889, 899, 916, 917, 918, 919, 922, 1019, 1026, 1035, 1044, 1045, 1046, 1047	To inhibit angiogenesis and reduce plaque formation
**Therapeutic Peptide Role: Agonist**			
Adenosine A2A Receptor	8FYN [62]	85, 168, 169, 177, 246, 249, 250, 253, 264, 267, 270, 274	To promote vasodilation and reduce inflammation
Apelin Receptor (APJ)	5VBL [31]	20, 21, 22, 23, 110, 114, 160, 163, 164, 175, 183, 198, 201, 202, 268, 271, 291, 1006, 1009, 1039, 1042	To improve cardiovascular function
Calcium-Sensing Receptor (CaSR)	7M3F [63]	20, 21, 22, 23, 24, 41, 54, 57, 59, 60, 98, 101, 106, 109, 112, 116, 117, 119, 132, 350	To promote vasodilation and calcium homeostasis
Glucagon-Like Peptide-1 Receptor (GLP-1R)	7RTB [64]	29, 42, 65, 75, 85, 96, 120, 137, 144, 152, 190, 197, 214, 233, 298, 372, 391	To improve glucose metabolism and reduce CAD risk
Low-Density Lipoprotein Receptor (LDLR)	2FCW [65]	86, 87, 88, 89, 90, 98, 105, 108, 110, 112, 118, 119, 129, 133, 135, 137, 142, 144, 147, 149, 151, 153, 157, 158	To enhance LDL clearance and reduce cholesterol levels

## Data Availability

The original contributions presented in this study are included in the article/Appendix A. Further inquiries can be directed to the corresponding author(s).

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
