# Peer review of "Evaluation of Structure Prediction and Molecular Docking Tools for Therapeutic Peptides in Clinical Use and Trials Targeting Coronary Artery Disease"

_ijms, 2025, doi:10.3390/ijms26020462_

Round 1
Reviewer 1 Report
Comments and Suggestions for Authors
The study provides valuable insights into the potential of therapeutic peptides for CAD and demonstrates the utility of advanced computational tools in drug design. However, some minor revisions will be required. The manuscript is generally well-written, but minor grammatical improvements are needed. Provide more details on how the peptides were selected for docking studies to clarify their relevance to CAD. Discuss the strengths and limitations of the tools (AlphaFold, I-TASSER, PEP-FOLD) and explain why these were chosen over others. Provide a brief explanation of the interpretation of binding free energies in the context of CAD therapy. Overall, the study offers valuable insights into CAD therapy, with minor revisions to enhance clarity and depth.
Comments on the Quality of English Language
The manuscript is generally well-written, but minor grammatical improvements are needed.
Author Response
Comments 1: The study provides valuable insights into the potential of therapeutic peptides for CAD and demonstrates the utility of advanced computational tools in drug design. However, some minor revisions will be required. The manuscript is generally well-written, but minor grammatical improvements are needed.
Response 1: We appreciate this feedback and have performed a detailed grammatical review of the manuscript. Adjustments were made to sentence structure, grammar, and flow throughout the text to enhance readability and precision. A professional editing tool was also employed to ensure consistency and accuracy in the language.
Comments 2: Provide more details on how the peptides were selected for docking studies to clarify their relevance to CAD.
Response 2: Thank you for your feedback. We’ve elaborated on the selection criteria for peptides for docking studies. Changes made: 5.1. Selection of Therapeutic Peptides. The study focused on seven therapeutic peptides known for their clinical relevance to coronary artery disease (CAD), specifically those approved for clinical use or in advanced clinical trials. These peptides included FX06, Liraglutide, Exenatide, Nesiritide, Apelin, GLP-1, and Atrial Natriuretic Peptide (ANP). The selection criteria prioritized peptides with documented roles in modulating key pathophysiological mechanisms associated with CAD, such as vascular inflammation, endothelial dysfunction, cardiac remodelling, and blood pressure regulation. This ensured that the peptides studied had a direct and meaningful connection to CAD pathophysiology. A systematic approach was adopted to identify peptides for docking studies. First, clinical literature and database searches were conducted to compile a list of candidate pep-tides with known or investigational relevance to CAD. Emphasis was placed on pep-tides targeting critical pathways, including renin-angiotensin-aldosterone signalling, nitric oxide bioavailability, and inflammatory cytokine modulation. Next, peptides with demonstrated efficacy or potential in mitigating CAD-related complications were shortlisted, ensuring diverse mechanisms of action, such as receptor agonism, antagonism, or enzyme inhibition. The amino acid sequences of the selected peptides were retrieved from reliable public databases, including UniProt [48] and the Therapeutic Target Database (TTD) [49], ensuring the integrity, accuracy, and reliability of the input data for structural modelling and subsequent computational analyses. Before proceeding with structure prediction, the retrieved sequences were verified for completeness and correctness. These peptides served as a foundation for evaluating the performance of peptide structure prediction and docking tools in this research. The docking studies focused on key receptor interactions relevant to CAD, such as the APJ receptor for Apelin, β-adrenergic receptors for ANP, and GLP-1 receptors for Exenatide and GLP-1. The study aimed to bridge the gap between computational modelling and translational research by selecting peptides with established or investigational roles in CAD pathways. The selection also incorporated therapeutic diversity to evaluate the computational tools across varying structural and functional peptide classes. For example, FX06 is known for its anti-inflammatory effects, Liraglutide and Exenatide target glucose regulation and cardiometabolic health, while Nesiritide and ANP are natriuretic pep-tides involved in vasodilation and fluid balance. Table 1 summarizes the therapeutic peptides used in this study, their amino acid sequences, and the binding sites identified for receptor interactions.
Comments 3: Discuss the strengths and limitations of the tools (AlphaFold, I-TASSER, PEP-FOLD) and explain why these were chosen over others.
Response 3: To address the feedback, we’ve elaborated on the details of the selection criteria and strengths and limitations of the selected tools (AlphaFold, I-TASSER, PEP-FOLD). Changes made: 5.3. Selection of Computational Tools. Three advanced, web-based tools were utilized to predict the 3D structures of the selected peptides: AlphaFold 3, I-TASSER, and PEP-FOLD 4. These tools were chosen based on their complementary methodologies, proven accuracy in peptide structure prediction, and ease of use, ensuring a robust analysis across varying peptide characteristics. Each tool offered unique approaches suited to different peptide characteristics, ensuring accurate and reliable structure predictions. AlphaFold 3 leverages cutting-edge deep learning algorithms to predict highly accurate peptide and protein structures [24]. By analyzing sequence data and applying evolutionary insights, Al-phaFold 3 generates predictions with remarkable precision, even without experimental structural data. One of its key strengths lies in its ability to effectively model both small and large peptides, making it an essential resource for studying therapeutic peptides. However, its performance can be less reliable for highly disordered regions or sequences lacking evolutionary conservation, which are sometimes present in therapeutic peptides. This tool is particularly valuable for predicting peptide conformations, as it can model both small and large peptides with a high degree of accuracy, making it an essential resource in structural biology [65]. I-TASSER combines template-based modelling with ab initio methods, providing robust peptide predictions, especially when experimental data is limited. This hybrid approach allows I-TASSER to build reliable models by identifying homologous templates from existing databases and using them to predict the peptide’s 3D structure. For sequences that are less homologous to known structures, I-TASSER employs ab initio strategies to explore alternative folding pathways, ensuring comprehensive structural coverage and enhancing its reliability for peptide prediction [66, 67]. While this increases its reliability, the accuracy of predictions can decrease for peptides with highly flexible or unique conformations, as the tool relies partially on the availability of structural templates. Nevertheless, its ability to integrate multiple methodologies makes it a versatile option for diverse peptide studies. PEP-FOLD 4, specifically designed for short peptides, utilizes a fragment-based approach that assembles small segments of known peptide structures to generate potential conformations [68]. This method provides accurate 3D models by focusing on the local backbone geometry, which is crucial when working with therapeutic peptides that typically consist of fewer than 50 amino acids. PEP-FOLD 4 is particularly valuable for peptides involved in receptor binding and drug design, as it ensures reliable conformational predictions for short peptides [69, 70]. However, its reliance on predefined fragment libraries may limit its utility for longer or highly disordered peptides. Despite this limitation, PEP-FOLD 4 is particularly suited for therapeutic peptides, where receptor-binding conformations are critical.
Comments 4: Provide a brief explanation of the interpretation of binding free energies in the context of CAD therapy. Overall, the study offers valuable insights into CAD therapy, with minor revisions to enhance clarity and depth.
Response 4: We appreciate the reviewer’s insightful comment. Below is a brief explanation of the interpretation of binding free energies (ΔG binding) in the context of coronary artery disease (CAD) therapy: Changes made: 2.4. Molecular Mechanics/Poisson–Boltzmann Surface Area (MM/PBSA). Calculations Binding free energy (ΔGbinding) is a crucial parameter in evaluating the stability and affinity of therapeutic peptides when interacting with their target receptors. In the context of coronary artery disease (CAD) therapy, lower binding free energy values indicate stronger and more stable interactions, which are often correlated with higher therapeutic potential. These interactions are essential for modulating critical biological pathways, such as reducing inflammation, improving endothelial function, and promoting vascular remodeling. For instance, peptides with significantly negative ΔGbinding values, such as Apelin with VEGFR2 (−199.17kcal/mol) and APJ (−171.62kcal/mol), suggest high binding stability, potentially enhancing their effectiveness in CAD treatment. However, while binding free energy provides insight into interaction strength, it must be complemented by experimental validation to confirm therapeutic efficacy and safety. Moreover, the receptor's biological relevance to CAD pathology must be considered to ensure clinical applicability.
Reviewer 2 Report
Comments and Suggestions for Authors
This study focuses on addressing the limitations of current Coronary Artery Disease (CAD) treatments by evaluating peptide-based therapeutics, which target underlying molecular mechanisms such as inflammation, oxidative stress, and lipid metabolism. Computational tools, including AlphaFold 3 for structure prediction and HawDock 2.0 for molecular docking, were systematically assessed for their accuracy and efficiency in modeling peptide-receptor interactions. Among the peptides studied, Apelin demonstrated exceptional binding affinities, stability, and therapeutic potential for CAD by effectively interacting with cardiovascular receptors critical to disease progression. The findings emphasize the importance of advanced computational methods in optimizing peptide drug design, highlighting Apelin as a promising candidate for future CAD therapies. Further research should explore its clinical applications and pharmacological profiles to refine treatment strategies.
The authors should address these questions:
- What are the potential challenges in translating Apelin’s high binding affinities and stability from in silico findings to clinical applications?
- How do the computational predictions align with existing experimental data or known pharmacological properties of Apelin?
Author Response
Comments 1: What are the potential challenges in translating Apelin’s high binding affinities and stability from in silico findings to clinical applications?
Response 1: In response to the reviewer's comment, we acknowledge the potential challenges in translating Apelin’s high binding affinities and stability from in silico findings to clinical applications. We’ve updated the Clinical Implications section. Changes made: 4.2. Clinical Implications. Despite these limitations, the findings of this study have significant clinical implications for the management of CAD. With its superior binding affinities and dynamic stability, Apelin emerges as a promising therapeutic peptide for CAD, potentially offering a novel treatment strategy. The study's identification of Apelin’s interaction with multiple receptors involved in cardiovascular homeostasis positions it as a candidate for future clinical development. Its role in modulating the renin-angiotensin system, enhancing nitric oxide production, and reducing oxidative stress can provide therapeutic benefits beyond conventional treatments such as statins and ACE inhibitors [42, 43]. Moreover, Apelin’s demonstrated efficacy in modulating inflammation and promoting vasodilation aligns well with the pathophysiology of CAD, providing a new avenue for treatment [8]. The results from this study suggest that Apelin could be beneficial in not only managing but potentially preventing CAD in patients with risk factors such as hypertension and diabetes. The multifaceted role of Apelin in cardiovascular health may also help manage comorbid conditions, thereby improving overall cardiovascular outcomes. However, translating Apelin’s high binding affinities and stability from in silico findings to clinical applications presents several challenges. First, while in silico predictions suggest promising receptor binding, the actual pharmacokinetics, and biodistribution of Apelin in vivo could differ significantly. Peptides often face challenges related to stability, bioavailability, and rapid degradation in the body, which may hinder their effectiveness when administered clinically. Furthermore, potential immunogenicity, which may arise from introducing a peptide into the body, could lead to adverse reactions or reduced therapeutic efficacy over time. Additionally, variability in patient receptor profiles and the presence of other comorbidities might affect how Apelin interacts with its targets, requiring further investigation into personalized therapeutic approaches. Finally, Apelin-based therapies' long-term safety and efficacy must be evaluated through clinical trials before widespread use in CAD treatment. Additionally, the study’s findings could facilitate the design of personalized therapies. Apelin’s high binding affinity and stability profiles suggest that patients with specific receptor profiles could benefit most from Apelin-based therapies. Developing predictive biomarkers to identify such patients could lead to more targeted and individualized treatment regimens, maximizing therapeutic outcomes while minimizing side effects [44].
Comments 2: How do the computational predictions align with existing experimental data or known pharmacological properties of Apelin?
Response 2: In response to the reviewer’s comment, we have revised the discussion to explicitly highlight how the computational predictions of Apelin’s binding affinities align with existing experimental data and known pharmacological properties. Changes made: Discussion section 2nd and 3rd paragraphs. Additionally, Apelin’s high binding affinity indicates its strong interaction potential with its target receptors, suggesting it could modulate these pathways effectively to manage CAD. This is consistent with experimental data showing Apelin's ability to enhance endothelial function and reduce hypertension [29]. The alignment of Apelin’s pharmacological action with its binding affinities for these receptors implies that it could be beneficial in preventing and treating cardiovascular diseases. For instance, Apelin binding to receptors such as the AT1R (angiotensin II receptor type 1) and β1AR (beta-1 adrenergic receptor) facilitates vasodilation [30], which enhances blood flow and reduces blood pressure, a key factor in preventing and managing CAD. MD simulations offer valuable insights into the dynamic properties of therapeutic pep-tide-receptor complexes. The RMSD, RMSF, RoG, and hydrogen bonding data reveal these complexes' conformational flexibility and stability during simulation. Apelin consistently exhibits the lowest RMSD values (2.124 to 2.356 Å) across all complexes, indicating a stable and well-maintained conformation throughout the simulation. This stability is crucial for drug efficacy, suggesting that Apelin can maintain its binding structure without significant deviations over time [31, 32]. The lower RMSF values observed for Apelin (1.511 to 1.867 Å) also indicate less fluctuation within its binding site than other peptides, indicative of a more rigid binding configuration and better stability. These results are consistent with findings from molecular dynamics studies of pep-tide-receptor interactions, where peptides with lower RMSF are considered to form more stable complexes, thereby enhancing their therapeutic potential [33, 34]. Moreover, the higher number of hydrogen bonds Apelin forms with its receptors suggests more robust and stable interactions. Hydrogen bonds play a significant role in maintaining the structural integrity of protein-ligand complexes, and a higher number of such bonds enhances the binding affinity and stability of the therapeutic complex [35, 36]. Comparing Apelin with other therapeutic peptides such as Liraglutide, ANP, and Exenatide, it becomes evident that Apelin shows superior binding affinity and maintains more excellent stability in its complexes. For instance, Liraglutide and Exenatide, while effective for diabetes [37, 38], exhibit lower binding energies and weaker receptor interactions in the context of CAD. This comparison highlights the specific therapeutic advantages of Apelin in managing CAD. Previous studies have shown that Apelin modulates multiple signaling pathways, such as the renin-angiotensin system (RAS) and insulin signaling pathways [39, 40], which are crucial for cardiovascular health. Its dual action on these pathways likely contributes to its enhanced therapeutic profile compared to other peptides, which typically target only one signaling pathway. For instance, Apelin has been shown to upregulate endothelial nitric oxide synthase (eNOS), leading to vasodilation and improved blood flow, which is beneficial in managing hypertension and heart failure [41, 42].